# Changes in Metabolism as a Diagnostic Tool for Lung Cancer: Systematic Review

**DOI:** 10.3390/metabo12060545

**Published:** 2022-06-14

**Authors:** Hanne Mariën, Elien Derveaux, Karolien Vanhove, Peter Adriaensens, Michiel Thomeer, Liesbet Mesotten

**Affiliations:** 1Faculty of Medicine and Life Sciences, Hasselt University, Martelarenlaan 42, B-3500 Hasselt, Belgium; elien.derveaux@uhasselt.be (E.D.); karolien.vanhove@uhasselt.be (K.V.); michiel.thomeer@uhasselt.be (M.T.); 2Department of Respiratory Medicine, Algemeen Ziekenhuis Vesalius, B-3717 Tongeren, Belgium; 3Applied and Analytical Chemistry, Institute for Materials Research, Hasselt University, Agoralaan 1—Building D, B-3590 Diepenbeek, Belgium; peter.adriaensens@uhasselt.be; 4Department of Respiratory Medicine, Ziekenhuis Oost-Limburg, Schiepse Bos 6, B-3600 Genk, Belgium; 5Department of Nuclear Medicine, Ziekenhuis Oost-Limburg, Schiepse Bos 6, B-3600 Genk, Belgium

**Keywords:** lung cancer, metabolomics, metabolite profile

## Abstract

Lung cancer is the leading cause of cancer-related mortality worldwide, with five-year survival rates varying from 3–62%. Screening aims at early detection, but half of the patients are diagnosed in advanced stages, limiting therapeutic possibilities. Positron emission tomography-computed tomography (PET-CT) is an essential technique in lung cancer detection and staging, with a sensitivity reaching 96%. However, since elevated 18F-fluorodeoxyglucose (^18^F-FDG) uptake is not cancer-specific, PET-CT often fails to discriminate between malignant and non-malignant PET-positive hypermetabolic lesions, with a specificity of only 23%. Furthermore, discrimination between lung cancer types is still impossible without invasive procedures. High mortality and morbidity, low survival rates, and difficulties in early detection, staging, and typing of lung cancer motivate the search for biomarkers to improve the diagnostic process and life expectancy. Metabolomics has emerged as a valuable technique for these pitfalls. Over 150 metabolites have been associated with lung cancer, and several are consistent in their findings of alterations in specific metabolite concentrations. However, there is still more variability than consistency due to the lack of standardized patient cohorts and measurement protocols. This review summarizes the identified metabolic biomarkers for early diagnosis, staging, and typing and reinforces the need for biomarkers to predict disease progression and survival and to support treatment follow-up.

## 1. Introduction

With an estimated 1.8 million deaths and 2.2 million new cases in 2020, lung cancer is the leading cause of cancer-related mortality worldwide. Covering 11.4% of all diagnosed cancers, lung cancer is the second most commonly diagnosed form of cancer [1]. With an overall five-year survival rate varying from 3–62% depending on the stage and regional differences, lung cancer is still a substantial burden to life expectancy in every country. When non-small cell lung cancer (NSCLC) is diagnosed early (i.e., in stages I and II), the five-year survival rate is about 62% in females and 51% in males. Unfortunately, about half of the patients are diagnosed in a later stage of the disease (locoregional advanced stage III and metastatic stage IV), limiting therapeutic options and decreasing five-year survival rates to 3% for both genders [2,3]. 

Despite the advances in understanding of risk factors, development, and treatment of lung cancer, it remains the leading cause of cancer death. With smoking being the dominant risk factor, disease prevention mainly focusses on tobacco avoidance [4]. Nevertheless, certain lifestyle factors, such as exposure to hazardous chemical substances, also pose a major risk for lung cancer development. Therefore, protective measures for risk professions minimize the potential harm of chemical exposure. Additionally, a healthy diet and a physically active lifestyle are known to have a reductive effect on the likelihood of developing cancer [5].

Screening aims to detect lung cancer before clinical symptoms are present. Low-dose computed tomography (LDCT) has proven effective in identifying suspicious pulmonary nodules or focal lung lesions. The National Lung Screening Trial showed that LDCT screening could reduce lung cancer mortality in high-risk patients by 20% compared to radiography [6,7,8]. Nevertheless, the most critical shortcoming of LDCT is its inability to verify whether the detected lung lesions result from benign lung disease or a malignant disease process [8,9]. PET-CT is an essential technique in lung cancer detection and staging, where almost no lung lesion goes undetected since its sensitivity reaches 96% [6]. However, regardless of its high accuracy and sensitivity, elevated ^18^F-FDG uptake is not cancer-specific. Increased levels of ^18^F-FDG uptake can also be detected in benign lesions such as inflammation and tuberculosis [10,11]. Therefore, the finding of increased FDG uptake often causes uncertainty about a diagnosis and sometimes even false-positives due to misinterpretation [12]. Although PET-CT is essential in disease diagnosis and staging, there is currently no fixed measure to estimate the probability of malignancy of a hypermetabolic single pulmonary nodule (SPN) in case of doubt. Therefore, PET-CT often fails to discriminate between malignant and non-malignant PET-positive hypermetabolic SPNs with a specificity of only 23% [6]. 

High mortality and morbidity, low survival rates, and difficulties with early-stage diagnostics provide good motivation for the search for biomarkers to improve the early detection process of lung cancer and life expectancy. Although such biomarkers are a hot research topic currently, few are used in the clinic. Proteomics and genomics have been widely used to identify new molecular targets and improve patient care [13,14]. Although they marked the past decades by the rapid development of new technologies, the individual genetic variability and costs associated with proteomic and genomic analysis make it impossible to monitor all relevant disease processes [15]. Therefore, a complementary technique, independent of genetic variability, is indispensable to improve lung cancer detection, staging, and treatment. Metabolomics has emerged as a valuable alternative. Since metabolites are the end products of cellular processes, alterations in metabolism automatically induce changes in metabolite concentrations, altering the individual metabolite profile [16]. The metabolic phenotype or fingerprint consists of many variables representing a single or several metabolite concentrations. In recent years, metabolite profiling/phenotyping, so-called metabolomics, has been used to investigate metabolic changes in plasma associated with lung cancer.

Hanahan and Weinberg describe several biological processes and characteristics associated with tumor development as the Hallmarks of Cancer [17,18,19,20]. These complex processes include bypassing growth suppression, limitless replicative potential, angiogenesis induction, metastasis activation, resistance to cell death, and chronic and uncontrolled proliferation [17]. Two important hallmarks joined the list a decade later, i.e., the evasion of immune destruction and metabolic dysregulation [18]. Another decade later, Senga et al. updated the hallmarks and added the genetic de- and transdifferentiation capability of cancer cells, epigenetic dysregulation that affects gene expression, altered microbiome, and altered neuronal signaling [19]. Those four additional hallmarks were confirmed and further described by Hanahan [20]. Tumor cells can establish all these hallmarks by genetic mutations and changes in the tumor microenvironment, resulting in metabolic changes.

Evidence has shown that cancer cell metabolism differs from that of normal cells. Tumors are often faced with nutrient- and oxygen-poor surroundings and develop various nutrient-scavenging strategies to bypass these limitations. The major reprogramming of the cancer cell energy metabolism is essential to enable cell growth and proliferation continuity. Cancer metabolic alterations have been organized into several features, such as the upregulated glycolysis, glutaminolysis, and amino acid and fatty acid synthesis pathways [21]. Lastly, there is continuous communication between stromal cells and malignant cells by which their metabolic interactions create a dynamic microenvironment. Cells surrounding the cancer cells, such as cancer-associated fibroblasts and immune cells, contribute to the metabolic needs of the malignant cells [22,23]. 

Tumor cells reprogram their energy production mechanism by consuming more glucose than normal cells and upregulation of their glycolytic capability. Glucose is preferentially converted into lactate instead of diverting pyruvate into the tricarboxylic acid (TCA) cycle (Krebs cycle), even in normoxic conditions [24]. This fermentation of glucose in aerobic conditions is known as the “Warburg effect”, providing rapidly dividing tumor cells with adenosine triphosphate (ATP) and metabolic intermediates needed to synthesize cellular components, such as DNA and proteins [25,26]. The excess lactate is secreted and accumulates in the extracellular space, the tumor microenvironment (TME), promoting the emergence of an immune-permissive microenvironment by attenuating T-cell activation [27]. Furthermore, excessive lactate stimulates angiogenesis by inducing VEGF secretion, an angiogenic factor, from tumor-associated stromal cells [28].

Additionally, cancer cells upregulate their glutaminolysis, which contributes to cell growth by promoting proliferation and inhibiting cell death. Glutamine is converted into α-ketoglutarate and used in the TCA cycle to provide intermediary metabolites, providing nitrogen and carbon skeletons to synthesize proteins such as amino acids [29,30,31]. Overfilling of the TCA cycle goes hand in hand with reactive oxygen species (ROS) production, which stimulates cell signaling and promotes cancer progression and spread. Tight control is crucial, as high levels of ROS are lethal to the cells [32]. Additionally, the pentose phosphate pathway (PPP) undergoes an upregulation in cancer cells since it plays a pivotal role in the nucleic acid synthesis and fatty acid synthesis by generating pentose phosphate and nicotinamide adenine dinucleotide phosphate (NADPH), respectively. The pathway is essential in cancer cell growth and survival under stress conditions [33]. Cancer cells are also characterized by a dramatic increase in lipid production, which is advantageous in forming lipid bilayers, and an increase in oxidative-damage-resistant saturated fatty acids [34]. 

It is well known that tumor cells mirror inflammatory conditions. Historically, such immune responses were thought to reflect an attempt by the immune system to attack malignant cells. That is not entirely untrue, but studies recently revealed that the inflammatory effect enhances tumor progression, besides the tumor-antagonizing effects. Inflammation contributes to several hallmarks by supplying the TME with growth factors for proliferative signaling, survival factors limiting cell death, proangiogenic factors, and enzymes facilitating invasion and metastasis [18,35]. 

These modified biological processes give rise to alternations in metabolite concentrations in tissues and biofluids (such as plasma, urine, bronchial aspirate, etc.). More than 150 metabolites have been identified in the altered lung cancer cell metabolism. This review provides an overview of studies that classified various metabolites using plasma, serum, or tissue samples with the possibility to aid clinicians in the differentiation between (1) lung cancer patients vs. healthy controls, (2) lung cancer patients vs. other cancer patients, (3) lung cancer vs. benign lung disease, (4) early-stage vs. advanced stage lung cancer, (5) lung cancer tissue vs. normal lung tissue, and (6) different histologies of lung cancer, as graphically shown in Figure 1. Some results are consistent in different published studies regarding specific metabolites. Therefore, an overview of the main study results is summarized in Table 1. However, there is still much variability between different studies about the used techniques and patient cohorts involved. Therefore, the crucial characteristics of the studies described in this review are compared in Table 2.

## 2. Methods

This systematic review was written following the 2020 updated Preferred Reporting Items for Systematic Reviews and Meta-analyses (PRISMA) guidelines [36,37]. The review protocol was submitted to the Prospero database (registration number 331945). PubMed was used as a database for the main search for articles. Covering the past 15 years, all articles were considered potentially useful if they covered the topics of lung cancer and metabolomics in plasma/serum or tissue samples. All articles were screened in which a comparison was made between all types of lung cancer and healthy controls, other cancer types, and patients with benign pulmonary disease, and a comparison was made between different stages and histologies of lung cancer. Metabolic alterations were documented to identify possible biomarkers that could facilitate lung cancer diagnosis, staging, and typing. 

The literature search is presented in Figure 2 using the PRISMA diagram available from http://prisma-statement.org/prismastatement/flowdiagram.aspx (accessed on 17 April 2022). The literature search in PubMed allowed the identification of 569 studies, of which 65 were included in the full-text evaluation. After a full-text read, 50 publications were excluded for various reasons, such as different focus (e.g., urine) or the inability to retrieve full-text, and 7 articles were added after checking the literature list of the already included articles. As a result, 22 articles were included in this review. For each included study, the characteristics of the study design, patients and their disease, and measurements techniques were registered. Finally, a summary was made of those characteristics, the identified metabolites, and their link to cancer metabolism. 

## 3. Results

### 3.1. Metabolic Differentiation between Lung Cancer Patients and Healthy Controls

Many studies have evaluated metabolic variations to differentiate between lung cancer (LC) patients and healthy individuals, primarily in serum and plasma samples, with high sensitivity and specificity. Several metabolites seem to have discriminative potential and might be clinically valuable as biomarkers.

However, inconsistent findings are reported regarding metabolites involved in the glycolysis and glutaminolysis. Glucose and lactate are both metabolites that, not surprisingly, stand out in many articles. Decreases in glucose and increases in lactate levels in serum samples of LC patients compared to healthy controls could represent the maintenance of increased aerobic glycolysis in cancer cells (Warburg effect) [25,26]. Zhang et al. [38] succeeded in discriminating between 25 early-stage LC patients and matched healthy controls, using proton nuclear magnetic resonance (^1^H-NMR) spectroscopy, with a 100% sensitivity and specificity. These investigators reported decreased glucose levels and increased lactate levels in LC serum samples compared to healthy controls [38]. Increased serum lactate levels were also reported by Puchades-Carrasco et al. [39] and Berker et al. [40]. The opposite finding, increased glucose and decreased lactate in LC vs. control, may be explained by the compensatory upregulated gluconeogenesis, using lactate, enabling the highly glycolytic character of cancer cells [26]. Louis et al. [41] reported increased plasma glucose levels and decreased lactate levels in a study with 98 LC patients and 89 controls [41]. Their findings are supported by Derveaux et al. [42]. 

Glutamine is another essential metabolite that has been extensively studied in cancer research. Additionally, for this metabolite, results are contradictory throughout several studies. Increases in glutamine and glutamate levels in LC serum compared to healthy controls confirm the increased glutaminolysis in cancer patients, supporting the production of metabolic intermediates for protein and nucleotide synthesis, essential for cell proliferation and survival [30,31]. Another possible explanation for the increased glutamine and glutamate serum levels may be the increased muscle protein breakdown seen in pathological conditions such as cancer. Proinflammatory cytokines, TNF-α and IL-6, mediate the augmented muscle protein degradation in cancer [43]. Zhang et al. [38] found increased glutamine and glutamate levels in LC serum samples compared to control samples, confirming increased glutaminolysis in cancer patients [30,31]. However, Puchades-Carrasco et al. [39] reported decreased glutamine levels in NSCLC serum samples. The targeted study group could explain differences, i.e., early stages of LC in the study of Zhang et al. [38] vs. all stages of LC in Puchades-Carrasco et al. [39]. This difference in the targeted population suggests that glutamine dependency increases as LC progresses, resulting in lower serum glutamine levels in the patients of all-stage study [31]. Louis et al. [41] detected increased plasma glutamine levels, despite including all stages of LC. It is worth mentioning that results may vary due to the different biofluids, i.e., plasma and serum, being used across different studies. 

Changes in the TCA cycle intermediates are often altered in lung cancer samples, such as decreased citrate and increased fumarate levels [44,45]. Decreased serum citrate in LC patients confirms the highly proliferative character of cancer cells since citrate is the primary substrate in the production of fatty acids and cholesterol, building blocks necessary for the increased proliferation and membrane biogenesis [46]. Fumarate has been associated with overcoming hypoxia by inhibiting the degradation of hypoxia-inducible factor (HIF) in cancer cells. HIF activity in tumoral hypoxia mediates angiogenesis, invasion, and metastasis by inducing glycolytic enzymes [47].

Most studies are consistent in the increase in leucine and isoleucine serum/plasma levels in cancer patients [38,39,41,42]. Branched-chain amino acids (BCAAs), including leucine and isoleucine, can regulate protein and lipid signaling pathways and cell growth. The upregulation of those amino acids could be explained by the tumor’s increased energetic and proliferative needs [48]. An attempt to counteract increased muscle protein breakdown in cancer patients might also explain the increased leucine levels. Leucine might counteract muscle degradation by stimulating protein synthesis by enhancing sensitivity for insulin [49].

Besides these metabolites returning in almost every paper on lung cancer metabolomics, there are papers describing other, less frequent metabolites associated with lung cancer. For example, ornithine [50,51,52] and arginine [52,53] are potential discriminative biomarkers for lung cancer. Ornithine aminotransferase, synthesizing proline from ornithine, promotes proliferation and metastasis of NSCLC by the upregulation of the miR-21 gene [54]. This finding suggests that proline and ornithine could be possible biomarkers for LC [52]. In addition, several studies observed that arginine could become limiting in states of rapid growth, such as malignancy [52,53]. Arginine is an essential amino acid for cellular growth and protein biosynthesis. Its overuse for these upregulated tumor processes and the inability of lung cancer cells to express argininosuccinate synthase (AS), the enzyme that regulates the biosynthesis of arginine, explains the decreased serum levels [53,55].

Increasing serum levels of tyrosine and histidine, precursors of catecholamine neurotransmitters and histamine, respectively, are mentioned in several studies [38,41,52]. Furthermore, tyrosine promotes lipid metabolism [56], whereas histamine is involved in cell proliferation and differentiation and regulates immune response [57].

Many papers describe a reduced level of plasma lipids in LC plasma/serum samples [42,58]. This finding is in accordance with the dysregulation of lipid metabolism in cancer [59,60]. Decreases in phosphatidylcholines are often seen in lung cancer patients, as they are major components of cell membranes [45,61]. Subsequently, serum levels of choline are decreased as it is the precursor of phospholipids in the membrane [39]. Unsaturated lipids and (very) low-density lipoproteins are also decreased due to the excess need for growth, proliferation, and metastasis [38,39].

Additionally, decreases in fatty acids, such as palmitic acid, support cancer cells’ high energy and biomass needs [50]. Alterations in ketone body levels are also consistent in different studies. Serum levels of ketone bodies, β-hydroxybutyrate and acetoacetate, have increased in LC patients vs. healthy individuals [38,39,44]. This increase can be observed when acetyl-CoA, derived from fatty acid oxidation, exceeds the TCA cycle [62]. This finding could also explain the reduced lipid levels in the serum of cancer patients.

### 3.2. Metabolic Differentiation between Lung Cancer Patients and Other Cancer Patients

Louis et al. succeeded in classifying 60 breast cancer (BC) patients and 81 LC patients with a sensitivity of 89% and a specificity of 82% [63]. They discovered that LC cells seem to be metabolically more active since they show a higher ^18^F-FDG uptake on a PET-CT scan. Increased glucose levels and decreased lactate levels suggest a more pronounced body response to the “Warburg” effect in LC patients. The difference in metabolic activity between lung tumor and breast tumor tissue is supported by the significant difference in membrane phospholipid concentrations in blood plasma samples from both patient cohorts. Decreased levels of sphingomyelin, phosphatidylcholine, and phospholipids with short, unsaturated fatty acid chains and increased phospholipids with long, saturated fatty acid chains were observed in LC samples compared to BC samples [63]. The difference in long saturated chains suggests that lung cancer cells have a more rigid membrane and are less sensitive to peroxidation [64]. The same research group extended the research by adding a group of colorectal cancer patients. They successfully classified 78% of the colorectal cancers, 95% of the breast cancers, and 84% of the lung cancers correctly. Further research needs to uncover the key metabolites in this discrimination [65].

To our knowledge, there are no other studies discriminating between primary LC versus any other primary tumors. However, Christen et al. published results of a study comparing primary BC to BC-derived lung metastases. Using ^13^C tracer analysis, they showed a more pyruvate carboxylase-dependent anaplerosis in BC-derived lung metastases rather than metabolizing glutamine to α-ketoglutarate to refill the TCA cycle, as seen in primary BC tissue [66].

### 3.3. Metabolic Differentiation between Lung Cancer and Benign Lung Disease

As mentioned earlier, PET-CT has high accuracy and sensitivity in detecting lung cancer. Nevertheless, it often fails to discriminate between cancer and benign lung lesions, such as inflammation and tuberculosis, since elevated ^18^F-FDG uptake is not cancer-specific [10,11]. Nevertheless, several studies have identified metabolites that can distinguish between patients with LC and patients with inflammatory lung conditions. For example, Deja et al. compared serum samples from NSCLC patients to those of chronic obstructive pulmonary disorder (COPD) patients. NSCLC patients showed decreased serum acetate, citrate, and methanol and increased leucine, choline, and ketone bodies compared to COPD patients [67]. 

Additionally, increased isoleucine, valine, lactate, and creatinine levels and decreased glutamine levels could discriminate between early-stage NSCLC compared to COPD [67]. Less discriminative potential was present when comparing advanced stage LC to COPD patients, presumably because advanced LC and COPD have a comparable level of tissue degradation [67]. As mentioned earlier, leucine was significantly increased in LC patients compared to healthy controls [38,39,41]. Interestingly, leucine levels were significantly decreased when comparing COPD patients to healthy controls [68,69]. This finding could make leucine a promising biomarker to distinguish between COPD and early-stage LC.

Puchades-Carrasco et al. [39] found that patients with benign pulmonary disease (BPD) have a different metabolite profile than patients with LC or healthy individuals. BPD is characterized by significantly higher levels of methanol, choline, and (very-) low-density lipoproteins ((V)LDL) and lower levels of glucose and lactate compared to LC [39], which is partly in accordance with the findings of Deja et al. [67]. A combination of five metabolites, including lactate and methanol, is presented that can discriminate between healthy controls, patients with BPD, and LC with 77% sensitivity and 78% specificity [39].

Lastly, Vanhove et al. [6] succeeded in presenting a model that differentiated between cancer and inflammation with 89% sensitivity and 87% specificity. In this model, tyrosine, glutamate, methionine, alanine, isoleucine, and lysine showed the most discriminative power. Moreover, glutamate was identified as a single diagnostic marker to discriminate between lung cancer and inflammation with 85% sensitivity and 81% specificity and an area under the curve of 0.88 [6].

### 3.4. Metabolic Differentiation between Early-Stage and Advanced-Stage Lung Cancer

Several studies identified differentially expressed metabolites in early-stage (I and II) and advanced stage (III and IV) LC. As mentioned earlier, Puchades-Carrasco et al. [39] documented decreased glucose levels and increased lactate levels in LC patients compared to controls. They observed that these serum changes become more prominent in advanced-stage LC compared to early-stage patient samples [39]. Parallel, higher levels of (iso)leucine and glutamate and lower levels of glutamine are observed in advanced-stage LC compared to early-stage LC. Comparable results were reported in the differentiation between healthy individuals and LC patients. The same significant differences between early and advanced stage LC support the higher energy and resource needs during cancer progression and metastasis [39].

However, Deja et al. [67] reported contrary results. They observed reduced levels of BCAA, lactate, and ketone bodies. The most significant difference between early- and advanced-stage LC, according to Deja et al. [67], was related to glycerol. In advanced patients, increased serum glycerol levels suggest lipid degradation or cell membrane rearrangement [67].

Berker et al. reported decreased serum glutamine and lactate levels in advanced stage lung cancer compared to stage I adenocarcinoma [40]. Higher energy needs in advanced LC could explain these findings. As mentioned before, glutamine and lactate are essential elements in the upregulated glutaminolysis and compensatory upregulated gluconeogenesis.

### 3.5. Metabolic Differentiation between Lung Cancer Tissue and Normal Lung Tissue

Many studies focused on LC metabolomics in biofluids. However, a limited number of studies applied it to tissue, presumably due to the highly invasive collection procedure [70]. Two recent studies observed an apparent metabolic alteration between LC and non-malignant tissue [71,72]. Some metabolites with a high predictive capability were identified. Moreno et al. detected changes in different glycolysis metabolites, such as a decrease in glucose and an increase in lactate and pyruvate [71]. In addition, intermediates of the TCA cycle, fumarate, and malate were accumulated in lung tumor tissue. Additionally, the PPP was altered in LC tissue, with a significant increase in ribose, ribose 5-phosphate, and fructose. Similar to biofluid studies [50,52,53], this tissue study by Moreno et al. [71] evaluated changes in arginine and ornithine levels in lung tumor tissue. Furthermore, the findings concerning fatty acid metabolism are comparable with biofluid studies [63]. Lung tumor tissue showed a significant increase in glycerol and long-chain fatty acids, supporting the rigid membrane structure of lung tumor cells and disrupted lipid peroxidation [71]. This finding was confirmed by Kowalczyk et al. [72].

### 3.6. Metabolic Differentiation between Different Histologies of Lung Cancer

Moreno et al. [71] evaluated the difference between lung tumor tissue and normal lung tissue and investigated whether adenocarcinoma (AC) tissue and squamous cell carcinoma (SCC) tissue present a different metabolic phenotype. A decrease in glycolytic metabolites was observed in both tumor subtypes, but a significant increase in lactate and pyruvate was only seen in SCC. All differences detected between LC and normal lung tissue were more prominent in SCC than in AC. Differences in the levels of metabolites in AC vs. SCC revealed that both subtypes regulate their cancer metabolism slightly differently. Nucleotide metabolism varies significantly for both histological subtypes, shown by a decrease in guanine in SCC [71]. A study by Kowalczyk et al. showed, slightly contrary to the study of Moreno et al. [71], that metabolites belonging to fatty acid and amino acid pathways were more upregulated in AC tissue compared to SCC tissue [72]. Both studies reported increased creatinine levels in SCC tissue vs. AC tissue [71,72].

Berker et al. described increased serum glutamine and glutamate levels in SCC compared to AC and decreased lactate levels when comparing stage I SCC to stage I AC. In addition, tissue analysis of both histological subtypes revealed an increase in alanine, valine, and lipid levels in SCC compared to AC. Additionally, tissue glutamate levels were higher in SCC than in AC, similar to the results of serum analysis [40].

**Table 1 metabolites-12-00545-t001:** Summary of most extensively studied metabolites and their alterations in lung cancer.

Involved Pathway	Metabolite	Plasma/Serum	Tissue
Healthy	BC	BPD	Early LC	AC	NLT	AC
LC	LC	LC	Advanced LC	SCC	LCT	SCC
Glycolysis	Glucose 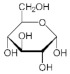	↓ [38,39]↑ [41,42]	↑ [41]	↓ [39]	↓ [39]		↓ [71,72]	
Lactate 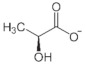	↑ [38,39,40]↓ [41,42]	↓ [41]	↑ [67]↓ [39]	↑ [39]↓ [40,67]	↓ [40]	↑ [71,72]	↑ [71]
Pyruvate 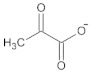						↑ [71,72]	↑ [71]
Glutaminolysis	Glutamine 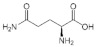	↑ [38,41]↓ [40]		↓ [67]	↓ [39,40]	↑ [40]		↑ [40]
Glutamate 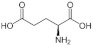	↑ [38,39]↓ [40]		↓ [6]	↑ [39]	↑ [40]		↑ [40]
BCAA metabolism	Leucine 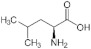	↑ [38,39,41,42]		↑ [67]	↑ [39]			
Isoleucine 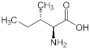	↑ [38,39,41,42]		↑ [67]	↑ [39]↓ [67]			
Valine 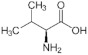	↑ [41]		↑ [67]	↓ [67]			↑ [40]
TCA cycle	Citrate 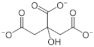	↓ [44]		↓ [67]				
Acetate 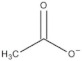			↓ [67]				
Fumarate 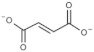	↑ [44,45]					↑ [71]	
Metabolism involving other amino acids	Tyrosine 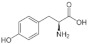	↑ [38,41,52]		↓ [6]				
Histidine 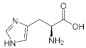	↑ [38]						
Urea cycle	Ornithine 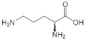	↓ [38,50,51,52]					↓ [71,72]	
Arginine 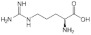	↓ [38,52,53]					↓ [71,72]	
Creatinine 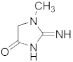			↑ [67]	↓ [67]			↑ [71,72]
Lipid metabolism	Choline 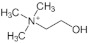	↓ [38,39]		↑ [39,67]				↑ [40]
(V)LDL	↓ [38]		↑ [39]				↑ [40]
Fatty acids	↓ [38]	↑ [41]				↑ [71,72]	↑ [40]
Glycerol 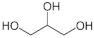	↑ [39]		↑ [67]	↑ [67]		↑ [71,72]	
Ketone bodiesβ-hydroxybutyrate 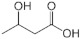 Acetoacetate 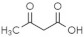	↑ [38,39,44]		↑ [67]	↓ [67]			

↑ indicates that the values are higher in the lower group compared to the upper group; ↓ indicates the opposite. For instance, the ↓ arrow for glucose means that LC samples presented lower glucose levels than the group of healthy controls. LC: lung cancer, BC: breast cancer, BPD: benign pulmonary disease, AC: adenocarcinoma, SCC: squamous cell carcinoma, NLT: normal lung tissue, LCT: lung cancer tissue, BCAA: branched-chain amino acids, TCA: tricarboxylic acid, (V)LDL: (very) low-density lipoproteins. References between brackets.

**Table 2 metabolites-12-00545-t002:** Characteristics of the studies most extensively described in this review.

Reference	Sample Type	Study Population	Measurement Technique	Statistical Analysis	Discriminative Capacity
Zhang et al., 2016 [38]	Serum	25 stage I LC 25 healthy controls	^1^H-NMR RRLC	OPLS-DA	LC vs. healthy: 100% sens, 100% spec
Puchades-Carrasco et al., 2016 [39]	Serum	90 A-LC 82 E-LC 27 BPD 87 healthy controls	^1^H-NMR	OPLS-DA	LC vs. healthy based on all metabolites: 92% sens, 95% spec, R² 0.931, Q² 0.873 LC vs. BDP vs. healthy based on 5 metabolites: 77% sens, 77.5% spec E-LC vs. A-LC: R² 0.779, Q² 0.592
Berker et al., 2019 [40]	Serum	Stage I LC: 27 SCC + 31 AC Advanced stage LC: 15 SCC + 20 AC 29 healthy controls	HRMAS-MRS	LDA CCA	ROC_AUC LC: 0.989
	Tissue	Stage I LC: 27 SCC + 31 AC Advanced stages: 15 SCC + 20 AC	HRMAS-MRS	LDA CCA	None reported
Louis et al., 2016 [41]	Plasma	Training: 233 LC vs. 226 healthy controls Validation: 98 LC vs. 89 controls 91 AC vs. 66 SCC	^1^H-NMR	OPLS-DA	Training LC vs. healthy: correct classification of 78% of LC, 92% of controls Validation LC vs. healthy: 71% sens, 81% spec AC vs. SCC: correct classification of 81% of AC, 38% of SCC
Derveaux et al., 2021 [42]	Plasma	Training: 80 LC vs. 80 healthy controls Validation: 34 LC vs. 38 controls	^1^H-NMR	OPLS-DA	Training LC vs. healthy: 85% sens, 93% spec Validation LC vs. healthy: 74% sens, 74% spec
Maeda et al., 2010 [52]	Plasma	141 LC vs. 423 healthy controls o 69 stage I, 72 advanced stage o 100 AC, 36 SCC	LC-MS	Logistic regression	ROC_AUC LC: 0.817 ROC_AUC stage I: 0.796 ROC_AUC AC: 0.795 ROC_AUC SCC: 0.860
Chen et al., 2015 [45]	Serum	30 LC (pre-op + post-op) 30 healthy controls	LC-MS GC-MS	PLS-DA	LC-MS: pre-op vs. healthy: R²X 0.527, R²Y 0.991, Q² 0.938 post-op vs. healthy: R²X 0.412, R²Y 0.992, Q² 0.935 pre-op vs. post-op: R²X 0.432, R²Y 0.906, Q² 0.975 GC-MS: pre-op vs. healthy: R²X 0.533, R²Y 0.854, Q² 0.747 post-op vs. healthy: R²X 0.518, R²Y 0.883, Q² 0.758 pre-op vs. post-op: R²X 0.457, R²Y 0.680, Q² 0.570
Deja et al., 2014 [67]	Serum	77 LC vs. 22 COPD o 17 E-LC + 60 A-LC	^1^H-NMR	OPLS-DA	COPD vs. LC: R²X 0.682, R²Y 0.762, Q² 0.568, AUC 0.993 COPD vs. E-LC: R²X 0.694, R²Y 0.809, Q² 0.651, AUC: 1 COPD vs. A-LC: R²X 0.663, R²Y 0.909, Q² 0.595, AUC; 1 E-LC vs. A-LC: R²X 0.732, R²Y 0.908, Q² 0.298, AUC: 0.904
Vanhove et al., 2018 [6]	Plasma	269 LC vs. 108 inflammation vs. 347 controls	^1^H-NMR	PLS-DA	LC vs. inflammation: based on all metabolites: 89% sens, 87% spec based on glutamate: 85% sens, 81% spec LC vs. control based on glutamate: 68% sens, 82% spec
Moreno et al., 2018 [71]	Tissue	68 LC and normal lung tissue of same patients o 33 AC vs. 35 SCC	LC-MS GC-MS	PLS-DA	None reported
Zhang et al., 2020 [44]	Plasma	156 stage I/II LC vs. 60 healthy controls	LC-MS HPLC-MS/MS	PLS-DA Logistic regression	Stage I/II vs. healthy: 0.919 sens, 0.900 spec, AUC 0.959
Kowalczyk et al., 2021 [72]	Plasma	72 LC vs. 20 COPD o 39 E-LC: 21 AC + 18 SCC o 33 A-LC: 11 AC + 15 SCC + 7 other	LC-MS: UHPLC combined with QTOF	PLS-DA	None reported
	Tissue	99 LC and normal lung tissue of same patients o 28 E-LC: 14 AC + 14 SCC o 71 A-LC: 19 AC + 40 SCC + 12 other	LC-MS: UHPLC combined with QTOF	PLS-DA	RPLC: AC vs. SCC vs. control: R² 0.983, Q² 0.853 HILIC: AC vs. SCC vs. control: R² 0.858, Q² 0.732
Qi et al., 2021 [50]	Plasma	98 LC vs. 75 healthy controls o 55 stage I/II+ 43 stage III/IV o 70 AC + 14 SCC + 14 other	LC-MS	Logistic regressionOPLS-DA	LC vs. healthy all stages RPLC: R²X 0.282, R²Y 0.960, Q² 0.703 HILIC: R²X 0.465, R²Y 0.962, Q² 0.820 Healthy vs. stage I/II vs. stage III/IV Top 5 significant metabolites: AUC 0.869, acc 0.829 Top 10 significant metabolites: AUC 0.947, acc 0.857 Top 20 significant metabolites: AUC 0.964, acc 0.900 Healthy vs. AC vs. SCC Top 20 significant metabolites: AUC 0.890, acc 0.830

LC: lung cancer, ^1^H-NMR: proton nuclear magnetic resonance spectroscopy, RRLC: rapid resolution liquid chromatography, (O)PLS-DA: (Orthogonal) Partial Least Squares Discriminant Analysis, A-LC: advanced-stage lung cancer, E-LC: early-stage lung cancer, BPD: benign pulmonary disease, sens: sensitivity, spec: specificity, SCC: squamous cell carcinoma, AC: adenocarcinoma, HRMAS-MRS: high-resolution magic angle spinning magnetic resonance spectroscopy, LDA: linear discriminant analysis, CCA: canonical correlation analysis, ROC: receiver operating characteristic, AUC: area under curve, (LC-)MS: (liquid chromatography–)mass spectrometry, GC-MS: gas chromatography mass spectrometry, pre-op: preoperative samples, post-op: postoperative samples, COPD: chronic obstructive pulmonary disorder, (U)HPLC: (ultra-) high-performance liquid chromatography, QTOF: quadrupole time of flight, RPLC: reversed-phase liquid chromatography, HILIC: hydrophilic interaction liquid chromatography, acc: accuracy. References between brackets.

## 4. Discussion

This review highlights potential plasma/serum metabolomic biomarkers associated with lung cancer. Table 1 shows an overview of the metabolites that were investigated in different studies as most discriminating in the differentiation between (1) lung cancer patients vs. healthy controls, (2) lung cancer patients vs. other cancer patients, (3) lung cancer vs. benign lung disease, (4) early-stage vs. advanced-stage lung cancer, (5) lung cancer tissue vs. normal lung tissue, and (6) different histologies of lung cancer. Several research groups have developed a metabolite signature for lung cancer. However, there is still considerable variability in study characteristics (Table 2) and results, making it difficult to draw a clear line to potential lung cancer biomarkers for early diagnosis, staging, and progression. Furthermore, it underlines the difficulties in translating research to clinical applicability.

The variability can be explained by the lack of uniformity and standardized procedures for sample collection and NMR analysis, the limited number of samples, and the variety of collected biofluids throughout all studies. The stability of the metabolites is critical for consistent results. Collection procedures and materials, time and temperature of storage, and sample processing protocols differ extensively and can all affect the stability and thus the concentration of the metabolites. Plasma is shown to be more stable than serum, for example. Another consideration regarding stability is the freezing-thawing cycles [73]. Additionally, researchers should consider the metabolite binding to plasma proteins, such as human serum albumin (HSA), which leads to an underestimated concentration of the bonded metabolites. An HSA-binding competitor, such as trimethylsilyl-2,2,3,3-tetradeuteropropionic acid (TSP), could be added to the samples to avoid unwanted metabolite binding, as described by Derveaux et al. [42].

Each analysis technique has its strengths and advantages on the one hand but, on the other hand, introduces a variety of metabolites that could be significant for lung cancer. The most intensively used techniques are MS and ^1^H-NMR. ^1^H-NMR is a quantitative tool and does not require extra sample preparation, such as derivatization. ^1^H-NMR is a fast technique with a low cost per sample compared to MS. MS-based metabolomics can analyze more metabolites and has better sensitivity than ^1^H-NMR, but ^1^H-NMR has higher reproducibility than MS and needs no solvent extraction steps in the sample preparation [74]. An extended list of the characteristics of the most commonly used measurement techniques mentioned in the studies included in this review is listed in Table 3.

Many studies focused on LC metabolomics in biofluids since biofluid samples are more easily accessible and convenient for investigation. In addition, relatively limited studies investigated tissue metabolomics, presumably due to the highly invasive collection procedure. However, biofluids are not organ-specific and reflect biochemical processes all over the body, complicating the interpretation of metabolomic results. Metabolomic tissue phenotyping serves a more straightforward interpretation since results originate at the localized site of the pathological process in a specific organ.

In addition to differences in experimental methods, it is necessary to consider patient-related factors when comparing the data obtained from various studies. The included patients differ widely across several studies, ranging from 25 LC patients and 25 controls [38] to 233 cancer patients and 226 controls [41]. Nevertheless, patient characteristics are even more critical since metabolites vary significantly with age, sex, dietary habits, smoking status, underlying diseases, medication intake, etc. The perfect biomarker is both sensitive and specific with diagnostic potential and clinical utility, independent of known predictors of the disease. The goal is to achieve superior performance to the standard of care-based strategy.

Potential biomarkers involve metabolites belonging to glycolysis, TCA cycle, PPP, urea cycle, other amino acid pathways, and lipid metabolism. As displayed in Table 1, metabolite changes vary in magnitude and direction depending on disease stage and type. These metabolites could aid the daily clinical practice in the early diagnosis and staging of lung cancer and provide molecular targets in developing new cancer therapies, which can considerably improve disease prognosis. The ability to distinguish between lung cancer and other cancers allows diagnosing whether LC originates from a primary lung tumor or as a metastasis from a tumor localized elsewhere. A metabolomic differentiation between LC and inflammation could support the results from PET-CT in diagnosing early-stage LC since ^18^F-FDG uptake is not cancer-specific. Increased levels of ^18^F-FDG uptake can also be detected in benign lesions, often causing doubt in diagnosis. Metabolomics could complement PET-CT to estimate the probability of malignancy of an SPN. The possibility of metabolic phenotyping to support the detection of LC in different stages and types could predict disease progression and survival. A correlation between stadium, aggressiveness, and the metabolic profile could support the choice of the most appropriate therapy and the follow-up of treatment response. Moreover, LC biomarkers could serve as a basis for personalized, targeted therapy since individual predictive biomarkers could improve efficacy and lower the toxicity of the treatment. The impact on patients would reach further than the advantages mentioned before in diagnosis, staging, and treatment follow-up. Healthcare costs could be reduced when LC is diagnosed early, avoiding extra hospitalizations and additional therapy, such as chemotherapy or radiotherapy. Additionally, metabolomics as a tool for early LC diagnosis could reduce emotional stress by providing a definite diagnosis within hours, whereas patients with doubtful SPNs are often advised to have another CT in three to six months.

Screening programs based on LDCT are already in place, and The National Lung Screening Trial showed that LDCT screening could reduce lung cancer mortality in high-risk patients by 20% compared to radiography [8]. In addition, screening and prevention programs could apply to high-risk patients with a 30-pack-year smoking history [75]. In the NELSON trial involving high-risk persons, lung cancer mortality was significantly lower among those who underwent volume CT screening than among those who underwent no screening. In this study, the cumulative rate ratio for death from lung cancer at 10 years was 0.76 (95% confidence interval [CI], 0.61 to 0.94; *p* = 0.01) in the screening group compared with the control group [76]. A blood-based test is more likely to encourage patients to participate in screening programs since sample collection is almost not invasive and straightforward. 

Metabolomics has been used to investigate the association between metabolites in pre-diagnostic serum and cancer risk. Kühn et al. [77] analyzed pre-diagnostic levels of 120 metabolites in 835 cancer cases. They report that higher levels of phosphatidylcholines were consistently related to a lower risk of breast, prostate, and colorectal cancer [77]. His et al. confirm the associations between specific metabolites (e.g., phosphatidylcholines) and pre-diagnostic breast cancer serum samples [78]. The prostate, lung, colorectal, and ovarian (PLCO) cancer screening trial was the first to report associations between pre-diagnostic serum metabolites and caffeine intake [79]. Despite these promising associations, it is a continuous struggle to unravel whether these metabolites play a direct role in tumorigenesis or are merely an early manifestation of disease in a preclinical state. 

Even more, several clinical trials are investigating the possibility of prediction of different cancer types with metabolomic analysis. Metabolomics urinalysis can play a role in the screening for colorectal cancer (CRC). Metabolites of the glycolysis, TCA cycle and urea cycle were identified as significant in the prediction of CRC, since increased urinary concentrations of these metabolites correlated with a more advanced stage of CRC [80]. A currently active trail in Taiwan (NCT03504098) investigates whether one-carbon nutrient intake can serve as a predictive marker for the development of lung cancer. Low folate intake is hypothesized to be associated with the increased risk for lung cancer, since it acts as a metabolic stressor. Another ongoing trial (NCT05185713) is analyzing vaginal metabolites to select biomarkers for the prediction of human papillomavirus-mediated cervical cancer and construct a cervical cancer risk and outcome prediction model. 

## 5. Future Directions

Metabolomics activities on lung cancer are still in the research stage, with too much variability between different studies. Therefore, there is an essential need for more extensive studies with extended patient cohorts and standardized measurement and analysis protocols to make lung cancer biomarkers clinically applicable. In addition, the lack of quantitative, reproducible results currently impedes the possibility of implementing LC biomarkers in daily clinical practice. Quantitative results that measure absolute metabolite concentrations would give an objective individual metabolic profile, making the results comparable between patients of different clinical centers. Furthermore, most studies focused on the differentiation between LC patients and healthy controls. These results are extremely valuable. Nevertheless, more studies are needed to identify the discriminative metabolites for LC stages and types to find biomarkers that could predict disease progression, survival, and support treatment choice and follow-up.

Once specific biomarkers are successfully validated in a clinical setting, the entire workflow, from sample collection to analysis, can be summarized in a Standard Operating Procedure (SOP), and HPLC, ^1^H-NMR, and MS assay kits can be developed to make the procedure easily repeatable and minimally subject to (pre)analytical errors.

## Figures and Tables

**Figure 1 metabolites-12-00545-f001:**
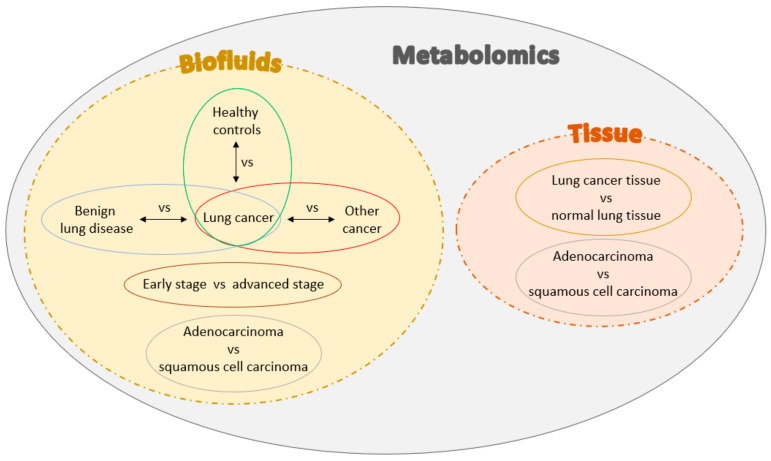
Types of lung cancer metabolite differentiation evaluated in this review.

**Figure 2 metabolites-12-00545-f002:**
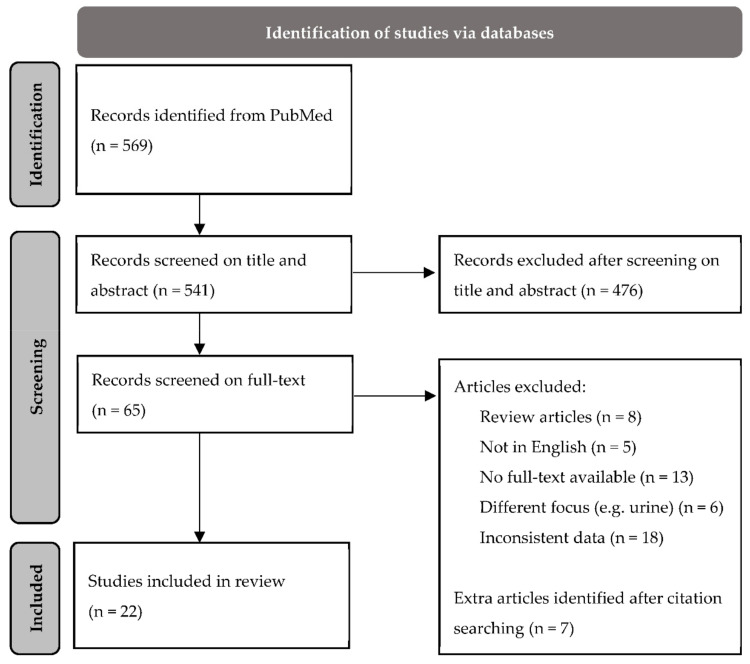
PRISMA 2020 flow diagram for new systematic reviews, which include searches of databases: flowchart of the literature process and selection of studies included in this review.

**Table 3 metabolites-12-00545-t003:** Characteristics of the measurement techniques used in the studies included in this review.

	^1^H-NMR	HPLC	(LC/GC)-MS
Sensitivity	Low	Higher	Highest
Sample preparation	Minimal sample preparation required	Extra sample preparation steps required: e.g., derivatization, solvent extraction	Extra sample preparation steps required: e.g., derivatization, solvent extraction
Number of detectable metabolites	30–100	300–1000+	300–1000+
Number of samples in one run	Analysis of 1 sample in 1 run	Analysis of more samples in 1 run	Analysis of more samples in 1 run
Cost per sample	Low	High	High
Reproducibility	High	Average	Average
Tissue samples	Can be analyzed directly	Requires tissue extraction	Requires tissue extraction
Speed	Fast	Slower	Slower

^1^H-NMR: proton nuclear magnetic resonance spectroscopy, (LC/GC-)MS: (liquid chromatography/gas chromatography –)mass spectrometry, HPLC: high-performance liquid chromatography.

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
