# Peer review of "Changes in Metabolism as a Diagnostic Tool for Lung Cancer: Systematic Review"

_metabolites, 2022, doi:10.3390/metabo12060545_

Round 1

Reviewer 1 Report

The paper refers to patients with diagnosis of breast cancer, which is one of the most common cancers resulting in high mortality. Early diagnosis in malignant tumors is a basic of effective therapy. These arguments make, that subject and work is very actual and useful in a clinical practice. Selection of literature is wide and very accurate, but suggest using more positions from literature in a discussion. After such an improvement I recommend publication of the paper presented for review.

Reviewer 2 Report

The present review is focused on metabolic differences in biological samples of patients affected by lung cancer, with the aim of identifying potential metabolic biomarkers for an early diagnosis and a correct classification. The paper is well-writing: the text is quite comprehensive and the topic is well summarized.

My unique objection is the plagiarism that I noticed in the writing of the paragraph 2. Methods (page 4-5, line 144-173). Indeed the authors of a systematic review with a similar topic, published exactly on Metabolites in 2021 (Metabolites 2021, 11(9), 630; https://doi.org/10.3390/metabo11090630), wrote the methods of their searching work using the same words and publishing the same flowchart in the figure 2 of Mariën and collegues’ review. I know that Madama et al.’s review was centered on non-small cell lung cancer (NSCLC), while this one considered all types of lung cancer patients, and I appreciated the choose of the author’s work to expand the cohort in order to distinguish their analysis from the review published one year ago. However, in my opinion, the originality of a work lies also in the writing of the scientific paper.

Moreover I suggest to authors to investigate if clinical screening or prevention trial about metabolomic analysis are actually opened or active on lung cancer and/or on other cancer types. Adding this information in the review could give it a more translationality increasing the interest of the readers.

I suggest the acceptance of this review after major revisions.

Reviewer 3 Report

1. The manuscript is a review article, authors could add a figure to show the srtuctures of the matabolites listed in Table 1.
2. Please also listed a table to comprae the advantages and disadvantages of the measurement techniques listed in Table 2.
3. Authors may a section to explain how to prevent the lung cancer.

Round 2

Reviewer 2 Report

I read the author's response and I appreciated the effort re-writing the paragraph 2. Methods. However, concerning my request to integrate the manuscript of screening programs based on metabolomics, I  have to highlight that numerous clinical trials are actually active for predicting different cancer type with metabolomic analysis and one is specific for lung cancer (NCT03504098). These results were emerged by a hurried research on clinical trial database, I would have preferred a deeper investigation by the authors in order to give a more clinical interest to their work.

Neverthless, in my opinion this review could be published after a minor revision.
